# The Price to Pay for Being Yourself: Experiences of Microaggressions among Non-Binary and Genderqueer (NBGQ) Youth

**DOI:** 10.3390/healthcare11050742

**Published:** 2023-03-03

**Authors:** Quinn Arijs, Aisa Burgwal, Jara Van Wiele, Joz Motmans

**Affiliations:** 1Transgender Infopunt, Ghent University Hospital, 9000 Ghent, Belgium; 2Center for Sexology and Gender, Ghent University Hospital, 9000 Ghent, Belgium

**Keywords:** NBGQ, non-binary, genderqueer, microaggressions, gender expression

## Abstract

This study explores the experiences of NBGQ youth with microaggressions. It investigates the types of microaggressions they face and their subsequent needs and coping mechanisms and the impacts on their lives. Semi-structured interviews with ten NBGQ youth in Belgium were conducted and analyzed using a thematic approach. The results showed that experiences of microaggressions were centered around denial. The most common ways to cope were finding acceptance from (queer) friends and therapists, engaging in a conversation with the aggressor, and rationalizing and empathizing with the aggressor, leading to self-blame and normalization of the experiences. Microaggressions were experienced as exhausting, which influenced the extent to which the NBGQ individuals wanted to explain themselves to others. Furthermore, the study shows an interaction between microaggressions and gender expression, in which gender expression is seen as a motive for microaggressions and microaggressions have an impact on the gender expression of NBGQ youth.

## 1. Introduction

A transgender* person is a person who does not identify with their assigned sex at birth, and the term is used to describe people who do not identify as cisgender [1]. Transgender* includes binary (trans man/woman) and non-binary or genderqueer identities. The terms “trans” or “transgender” often get used in a confusing manner to signify trans men and women, while the transgender* umbrella includes a diversity of identities, such as non-binary, genderqueer, agender, genderfluid, polygender, and more [2]. When “trans” is used in research to mean trans men and women or the general trans* umbrella is used, it is often not clear what is being referred to. For this reason, we use the term “trans*”, in which the asterisk signifies that it is used as an umbrella term to emphasize inclusion of a range of gender identities [3,4].

Non-binary and genderqueer identities comprise multiple gender identities that are not (merely or completely) male or female and fall outside the gender binary [5]. Richards et al. [5] state that people with non-binary or genderqueer identities may identify as both male and female at one time, as different genders at different times, or as no gender at all, or they may dispute the very idea of there being only two genders. Non-binary and genderqueer are used as umbrella terms for various identities, such as pangender (a person whose gender identity reflects multiple gender identities [6]), agender (a person who experiences a lack of gender or does not feel any connection to masculinity or femininity [7]), or genderfluid (people who experience their gender identity as shifting and changing [7]). This list is not exhaustive but gives some examples of what a non-binary or genderqueer identity can mean. For this research, non-binary and genderqueer were defined as people who do not identify as (only or entirely) a man or woman. This is based on self-identification.

To talk about identities under the trans* umbrella that are not binary, we use the term “non-binary and genderqueer” (NBGQ) (except when referring to the work of other authors; in this case, their own terminology is used). While “genderqueer” is associated with political statements against the gender system and “non-binary” is a more apolitical statement of identity, both have been used as umbrella terms to include a variety of gender identities [8]. In this paper, we chose to use both terms simultaneously, using the abbreviation NBGQ to emphasize the variety of gender identities and to avoid valuing one term over another. This abbreviation has been increasingly used in research over recent years [8].

According to the review by Zhang et al. [9], the proportions of people identifying as (binary) transgender are 0.3–0.5% among adults and 1.2–2.7% among children and adolescents. Including gender diversity, the proportions increase to 0.5–4.5% among adults and 2.5 to 8.4% among children and adolescents, which indicates a rise in the proportion of trans* and gender-diverse people [9]. Additionally, an online survey indicates a growing proportion of young trans* people (14–25 years old) identifying as non-binary [10]. In Belgium, the estimate for the proportion of gender-incongruent (=binary trans) people is 0.6–0.7% based on a representative population sample [11]. Additionally, Van Caenegem et al. [11] estimated the proportion of gender-ambivalent (=NBGQ) people to be 1.9–2.2%. In a Dutch study, 0.8–1.1% of people were reported to be gender-incongruent, while 3.2–4.6% of people were reported to be gender-ambivalent [12]. A study by the European Union Agency for Fundamental Rights [13], using a probability sample, estimated that 23% of European trans* people identify as non-binary, 9% as genderqueer, 11% as genderfluid, 7% as agender, and 1% as polygender. Another probability sample from Belgium in a study by Motmans et al. [14] estimated the amount of trans* people who identify as non-binary to be 25%. Lastly, in the Canadian census, among 30.5 million Canadians, 0.33% self-identified as transgender or non-binary [15].

### 1.1. Trans* People and Violence Experiences

Among European and Belgian lesbian, gay, bisexual, trans, and intersex people (LGBTI) (various umbrella terms are used to refer to lesbian, gay, bisexual, trans, and intersex people; in this article, multiple abbreviations are used, such as LGBT, LGBTI, LGBTI+, LGBTQ, and LGBTQIA+, as they were used by the original authors), 11% and 14%, respectively, have been attacked physically or sexually in the past 5 years [13]. For transgender people, these proportions amount to 17% in the EU and 19% in Belgium [13]. Up to 48% of transgender people have faced harassment and 50% in Belgium, which is the highest proportion among all subgroups [16]. Belgium has the second highest numbers of reported cases of people suffering non-verbal in-person harassment due to being LGBTI in the EU (28%), with a higher percentage (35%) for trans people [16].

Although trans men and women’s experiences with violence have been widely documented, NBGQ people’s experiences with violence have not gained much attention. Most existing research focuses on binary transgender people alone, on comparisons between binary and NBGQ transgender people, or on the general trans* population, making it difficult to ascertain the specific experiences and needs of non-binary and gender-diverse people [17]. Additionally, Aparicio-Garcia et al. [18] indicate that trans* people experience more violence than non-trans* people, with non-binary and gender-diverse people facing the worst outcomes and the highest numbers of traumatic experiences [17].

In Belgium, 5–9% of NBGQ people were found to have had experiences of physical or sexual violence in the year prior to the study due to being LGBTI [16]. The more subtle the experiences were, the more NBGQ people had experienced them: 23–32% of NBGQ people in Belgium experienced verbal in-person harassment and 22–49% non-verbal in-person harassment in the previous year due to being LGBTI [16]. Subtle forms occurred more often: 33–36% of the NBGQ people who had experienced offensive gestures and 21% who had experienced insults or name-calling had encountered these experiences more than six times in the previous year [16]. Additionally, Motmans et al. [19] indicate that most transgender people face verbal or psychological violence (80%), while a smaller proportion face sexual (32%), physical (27%), and material (18%) violence. However, research often focuses on physical and sexual violence, while verbal or psychological violence remain underexplored, possibly because they are harder to grasp or because people may not categorize these acts as “violent enough”.

### 1.2. Microaggressions

In this context, this research focused on more subtle verbal and psychological forms of violence. These experiences are often categorized under the term “microaggressions”, a concept that originated in racial discrimination research. Racial microaggressions were defined as verbal and physical racial attacks and covert behavioral and verbal comments that demean a person’s racial identity or invalidate minorities’ thoughts and experienced realities [20]. The concept has evolved over time and recently it has been applied to LGBT people [21]. Microaggressions refer to subtle, covert, everyday experiences of discrimination against marginalized groups that can be conscious or unconscious and can include behaviors and comments—both verbal and psychological—that invalidate someone’s identity or thoughts [21]. However, research on LGBT people and microaggressions has been focused on workplace settings [22,23] and experiences of genderqueer people and trans men and women, or genderqueer people and the general LGBTQ population have often been mixed [24].

A literature review by Nadal et al. [24] on microaggressions and the LGBTQ community found no publications on genderqueer or gender-nonconforming people and microaggressions. Most studies were undertaken on microaggressions and LGB people, the general LGBTQ population, and the general trans* population [24]. It is thus hard to deduce NBGQ people’s specific experiences and needs; therefore, a study on their experiences of microaggression is needed.

Three forms of microaggressions exist: micro-assaults (overt verbal or nonverbal insults and behaviors), micro-insults (statements or actions that demean a person’s identity), and microinvalidations (negating or nullifying minority groups’ psychological thoughts, feelings, or experiential reality) [25]. In addition, Nadal et al. [26] identified 12 themes in the experiences of microaggressions among trans* people: using transphobic or incorrectly gendered terminology, the assumption of a universal transgender experience, exoticization, discomfort around/disapproval of transgender experience, endorsement of gender-normative and binary culture or behavior, denial of the existence of transphobia, assumptions of pathology or abnormality, physical threats, denial of individual transphobia, denial of bodily privacy, familiar microaggressions, and systemic microaggressions regarding the use of public restrooms, the criminal justice system, healthcare, and identification documents.

For the purpose of this research, we defined microaggressions as forms of psychological and verbal violence manifesting as the more subtle and day-to-day experiences of violence, such as (but not limited to) misgendering, deadnaming, denial of identity, making someone feel ashamed of or bad about themselves, and inappropriate remarks or jokes. Due to the nature of microaggressions as being subtle and covert, not everyone may identify these occurrences as microaggressions. While these occurrences can be objectively defined as microaggressions, the subjective experience of these occurrences as microaggressions is different to different people. In this research, we only included microaggressions that were defined as such by participants; however, this does not mean that certain occurrences that are not defined as such could not still be categorized as microaggressions.

### 1.3. Additional Challenges of NBGQ People

NBGQ people face additional challenges when dealing with microaggression. For example, the research by Kiekens et al. [27] indicate a differences in the experiences of microaggressions among sexual minority and gender minority groups, with gender minority youth experiencing more familial microaggressions, invalidation of LGBTQ identity, and threatening behaviors.

#### 1.3.1. Misgendering, Deadnaming, and Authenticity

A typical form of microaggression against trans* people is minimalizing or denying the person’s experiences, such as misgendering or deadnaming [28]. Stating that the person is not a real man, woman, or NBGQ person counts as psychological abuse [28]. The research by Pulice-Farrow et al. [29] indicates differences in the experiences of binary trans and NBGQ people: binary trans people are misgendered more by being referred to with their gender assigned at birth while NBGQ people are misgendered more using binary language. The research by Pulice-Farrow et al. [29] also indicates that challenging authenticity differs between binary and NBGQ identities: binary trans people are questioned about not being real men or women, while NBGQ people are questioned about being real trans* people. Additionally, gender minority youth are more likely to experience invalidation of their identities or familial microaggressions [27].

#### 1.3.2. The Role of Gender Nonconformity in Experiences with Violence

Another factor is that NBGQ people are hardly able to pass, which means to be perceived and recognized as the gender they identify with, as there is “nothing to pass as” [30]. It is almost impossible for them to present in such a way that they will definitely not be misgendered [30]. More visible forms of “being different” often lead to more stressful situations and discrimination [31]. Pulice-Farrow et al. [29] indicate that microaggressions change according to the transitions one has undergone, and Gordon and Meyer [32] suggest that gender nonconformity is an incentive for prejudice and discrimination. Furthermore, Klemmer et al. [33] find that gender-nonconforming students are at higher risk of experiencing school violence than gender-conforming students, regardless of their gender identity or sexual orientation, which is in line with research by D’Haese et al. [34]. Additionally, NBGQ people experience less validation and more social pressure to conform to the norm [30]. Butler [35] states that gender norms are inherently violent as they have regulatory power that shapes how people can and should look and behave. Passing indeed often intersects with safety concerns, with self-presentation being weighed against maximal safety [30]. Furthermore, avoidance behavior is an coping strategy often applied among trans* people [14]. The study by the FRA [16] shows that, in Belgium, 5–14% of NBGQ people always and 12–41% often avoid certain locations for fear of being assaulted, threatened, or harassed. Additionally, NBGQ people avoid being open in public spaces (37–74%), around their family (18–33%), at the workplace (35–41%), at school (6–19%), and at bars or restaurants (18–32%) for fear of being assaulted, threatened, or harassed. Higher numbers of NBGQ people are not open about their identity with anyone in their family (25–45%) than trans men (16%) or trans women (1%) [16].

These findings indicate that gender expression, rather than sexual or gender identity itself, plays a role in the experience of violence. This may be related to the cisnormativity that is present in society. Collier and Daniel [36] describe cisnormativity as “the seemingly natural and ahistorical assumption of cisgender identities that structures institutions and interactions and results in the erasure of gender-variant experiences and realities”. Non-binary people are often seen as a threat to gender structures since they challenge the notion of gender as a fixed binary fact rooted in biology [37]. Among trans* people, experiencing violence is related to openness about their gender identity in everyday life (among family, friends, neighbors, at work, or when using health services), while for other LGBTI groups, the prevalence of violence remains the same regardless of openness [13]. Therefore, experiences of microaggressions need to gain focus, as well as how gender expression and nonconformity influence and interact with such experiences.

#### 1.3.3. Epistemic Exploitation

Another form of violence inflicted on minority groups is epistemic exploitation. Toole [38] coined the term “epistemic exploitation”, which she uses to refer to epistemic oppression in which marginalized knowers are expected to educate dominantly situated knowers about their oppression. She argues that this act is exploitative because it requires emotional and cognitive labor that is unrecognized and uncompensated and it places an unfair burden on those who are already marginalized. In this way, Toole [38] states, mental energy is diverted away from people’s own projects, goals, and interests in the service of someone else. Fiani and Han [30] have researched activism and educating others as a final stage of identity development. They find that most NBGQ people report that assuming an educator role requires demanding emotional labor, and often they prefer people to refer to the Internet instead [30].

### 1.4. Situation of NBGQ Youth

According to Arnett [39], the impact of stress from negative day-to-day experiences among minorities is especially pronounced for adolescents and young adults between 18–25 years of age since they often transition from living at home to living independently. Specifically, for NBGQ youth, the separation from their parents is a crucial step in the development of their gender identity and their being able to accept themselves independently of their family’s expectations [30]. In line with this, research on racial minorities and adolescence shows that young people are at increased risk of exposure to microaggressions because they begin spending their free time outside their home and in public spaces [40]. This—combined with the fact that the mean age of initial consciousness of being non-binary is 15.6, while the mean age of coming out is 25.8 [14]—means that young NBGQ people may experience more microaggressions from their close environment, since they often have not come out yet. Adolescents and young adults between 18 and 24 years old experience the most physical and sexual violence within the trans* population in the EU and have the highest rate of non-verbal in-person harassment due to being LGBTI in Belgium [16]. Lastly, an increasing number of young trans* people identify as non-binary [8]. For example, Statistics Canada [15] shows that proportions of trans* and NBGQ people are three to seven times higher for people born between 1981 and 2006 than for people born in 1980 or earlier. Therefore, this research focused on the experiences of NBGQ youth, who have been defined by studies as adolescents and young adults and are operationalized for this study as people aged 18–25.

### 1.5. Impact

Microaggressions can have a significant impact on NBGQ people’s lives. Most trans* people in Belgium who experience violence (57%) state that this had an impact on one or more areas, such as psychological problems or fear of going outside [16]. Carroll [41] labels the stress of living in a hostile environment with daily microaggressions “mundane extreme environmental stress”. This type of stress is experienced day-to-day, influences the psyche and world view, and is environmentally induced. According to the minority stress model, psychosocial stressors such as violence have negative effects on minority groups’ lives [31]. The type of stress that microaggressions based on gender identity elicit is extra strong, since it is out of one’s control. The impact of daily stress depends strongly on resources, networks, and coping strategies, which can lower the negative effect [31]. Therefore, it is important to explore whether NBGQ youth seek help after experiences with violence and what their needs in this regard are. According to attribution theory, attribution processes influence the outcomes of violent experiences [42]. Whether NBGQ people attribute the experience to themselves or to discrimination plays a significant role in the effects violence has on their wellbeing. This theory distinguishes between behavioral and categorical self-blame, respectively, when blaming oneself for something one did or for who one is, with the latter having more negative effects on wellbeing. Since gender identity is about who one is and gender expression about what one does, it is interesting to explore whether these processes play a role in the experience of violence among non-binary people.

The research by Jones et al. [43] shows that a lack of validation, which microaggressions may elicit, leads to worse mental health and quality of life among non-binary and transgender people. Sue [25] also highlights that more covert and difficult-to-identify microaggressions often have a stronger impact and a higher emotional toll than more overt encounters and posits that unintentional, covert forms of microaggressions may be extra dangerous because they often go unseen and are more pervasive.

Furthermore, a difference may exist between binary and NBGQ trans people in terms of the resources required to cope with negative experiences: non-binary people report community support and creative outlets as being more important than binary trans people [30].

### 1.6. The Belgian Context

The increasing attention paid to legal gender registration procedures makes Belgium an interesting place for this research. Additionally, in Belgium, only 51.3% of NBGQ people live according to their gender identity and 33.3% of NBGQ people are never or seldom approached and treated according to their gender identity [14]. Attitudes towards NBGQ people are not entirely supportive: 42% of Belgians do not support adding a third option as a gender marker on official documents [44]. Microaggressions are recurrent: regarding people who have experienced threatening gestures or inappropriate staring in the past year, Belgium has the lowest amount of NBGQ people in Europe who report experiencing this only once (as opposed to multiple times) [16]. This indicates that NBGQ people’s situation is especially challenging. In conclusion, while changes are occurring at the policy level to include NBGQ people, the population feels divided, which may lead NBGQ people to live in a not-so-supportive environment where microaggressions may be common. Various data about experiences of verbal in-person harassment in Belgium can be found, as described in Section 1.1. However, specific data about microaggressions among non-binary people in Belgium are not available.

### 1.7. Research Goals

This research aimed to provide insight into the experiences of NBGQ youth (18–25 years old) with microaggressions; the subtle and day-to-day experiences of verbal and psychological violence they face; and the subsequent needs, coping mechanisms, and impacts on their lives. To do this, we focused on distinct aspects of the topic. First, we aimed to show the experiences of NBGQ youth with microaggressions in general and, specifically, those related to being NBGQ (misgendering, deadnaming, claiming inauthenticity, and epistemic exploitation). Secondly, we explored how gender expression and nonconformity interact with these experiences. Lastly, we studied the needs NBGQ have when confronted with microaggressions and how they cope with these experiences.

## 2. Materials and Methods

### 2.1. Participants

Ten participants were interviewed. Seven identified as non-binary, one as genderfluid, and two did not label their identity. The youngest participant was 18 years old and the oldest was 24 years old. All participants were born in Belgium and were white, although one participant had a parent who migrated to Belgium. Nine participants lived in Flanders and one in Brussels. Seven participants had higher education or were studying in a higher education establishment. Five participants were students, three were unemployed, and two had jobs. Lastly, four participants lived alone or with a supporting partner or friends, two participants lived in student residences, one participant lived in a government facility, and three participants lived with their parents.

### 2.2. Procedure

The research was approved as part of the overarching study “Genoeg–Enough–Assez” by the Medical Ethics Committee of the Ghent University Hospital. As the goal was to gauge people’s subjective experiences, a qualitative approach was chosen. Data were collected using semi-structured interviews to explore experiences of microaggressions among NBGQ youth. This method provided focus and ensured complete and in-depth answers while leaving room for flexible input from participants. Semi-structured interviews were conducted using a topic list, which is described in Section 2.3.

Participants were recruited through the survey in the study “Genoeg–Enough–Assez”. The survey was promoted through posters, flyers, and social media in LGBTI+ and non-LGBTI+ contexts. At the end of the survey, people had the option to leave their contact information if they wished to be invited for an interview. Anyone living in Belgium for at least two years and identifying with an LGBTI+ label could participate, regardless of whether they had experiences of violence. Among the people 18 to 25 years old who responded to the question “How would you describe your gender identity at the moment” by ticking the box “gender diverse (genderqueer, non-binary, agender, genderfluid)” and decided to leave their contact information, a random selection was used to decide who would be contacted and invited. They were sent an e-mail with an invitation to the interview, and one reminder was sent if a participant did not reply. Thirteen participants were contacted; three did not respond to the invitation, which made the non-response level relatively low. In the end, ten participants were recruited, reaching the level of saturation. The participants were all unknown to the researchers.

Participants were invited for the interview at a quiet location of their choice. This location was generally in their hometown at a separate room booked in an LGBTQ+ space (such as a rainbow house), youth center, or university building, although one interview took place at a bar and two interviews took place in a park. The interviews were carried out by the first author. Only the participant and interviewer were present. Before the interview, the interviewer introduced themselves and shared their gender identity, affiliation, and why they were undertaking the research with the participant. The background and objectives of the research were explained, along with which topics the interview would contain. We defined microaggressions to them as we described in the introduction. They then read and signed an informed consent form. They were asked to give permission for an audio recording of the interview to be produced and the interviewer explained how the data would be treated and that their anonymity would be assured. All participants gave their consent to record, so all the interviews were audio-recorded. The interviews took approximately one hour. Due to the recording, no notes were taken during the interview. Participants received a debriefing after the interview with contact information for resources they could reach out to if they wanted to talk about their experiences. Participants did not receive an incentive to participate in this study.

After the interview, recordings were saved at a secure location under an alphabetical code. Then, the recordings of the interviews were transcribed in full, leaving out personal details participants shared that could lead to the possibility of them being identified through the data. The data were pseudonymized and transcriptions were saved under a different alphabetical code. Once the transcription was finished, the audio recording was deleted. Names used in Section 3 are fictional. Transcripts were not returned to participants for comment or correction.

### 2.3. Material

An interview guide was used to provide a structure for the semi-structured interviews. This guide included questions regarding the overarching violence study, as well as additional questions for this specific research. The topic list for the overarching study included questions about general experiences of violence, reporting violence, help-seeking, and attitudes about official instances. This was constructed to fit the goals and framework of the overarching study in collaboration with the guidance committee. It was then adapted to explicitly include experiences of microaggressions by applying the same questions to experiences of microaggressions instead of violence as a whole, and questions regarding the interaction with gender expression were added. This was approved by the supervisors.

### 2.4. Data Analysis

The transcriptions were analyzed using Braun and Clarke’s thematic analysis [45]. This is a method for identifying, analyzing, and reporting patterns and themes within data [45]. A theme is a patterned response that captures something important about the data in relation to the research question [45]. This analysis is a suitable approach for research questions that aim to explore people’s experiences. A semantic approach was used as opposed to a latent approach to reduce the subjectivity of the researcher’s judgment and because the interest of this study was more related to people’s stated experiences than their assumptions.

The thematic analysis was carried out by the first author using the six steps proposed by Braun and Clarke [45]:The data were transcribed and, during this process, initial notes or remarks were written down;Initial codes were generated using NVivo software. During this step, overarching codes were assigned deductively based on the research questions, such as “types of microaggressions”, “reaction”, “motive”, “impact”, “needs”, “coping”, and “gender expression”. Then, within these categories, descriptive codes were assigned inductively from the data to segments that were relevant to the research topic;Connections between codes were sought and codes were sorted into potential themes with the help of a visual thematic map;Themes were reviewed. Minor themes that did not have enough data to support them were scrapped, and themes were joined or broken down according to what seemed appropriate. This was reviewed by reading through the data segments within one theme to verify whether they formed a coherent pattern, and the dataset was considered as a whole to make sure the themes accurately represented the data;Then, the themes were defined, and we specified what each theme brought to the general analysis and in what ways they related to each other;Lastly, segments of the data were selected to represent and provide evidence for each theme and to explain how the themes answered the research questions.

As thematic analysis is interpretative and relies on the researcher’s judgment, it was important for the first author, who conducted the interviews and analyzed the data, to reflect on their positionality as a researcher and how their perception may have influenced results. They are a young NBGQ person, a master student at the time of the study, and barely older than the age group they were researching, which had multiple consequences. On the one hand, talking to an NBGQ person may have allowed the participants to feel safer, to be more open, and to talk more freely. Several participants indicated that this gave them the feeling they did not have to explain their identity and that they were talking to someone who could understand their experiences. However, it was important to be mindful that this may also have led to participants not explaining their experiences in as detailed a manner as they would to someone that they thought did not understand. To avoid this, it was important to not make assumptions and to keep asking questions. Furthermore, the researcher has their own experiences with NBGQ microaggressions and gender expression and their own needs and coping mechanisms. It was crucial to be aware that their individual experiences were personal, did not speak for all NBGQ experiences, and were not generalizable. Therefore, it was important that, during the interviews and analysis, they did not steer or interpret people’s words to reflect their own thoughts and experience. This was an active, conscious process of which they needed to be aware.

## 3. Results

Thematic analysis resulted in the identification of themes and subthemes. The identified themes and subthemes are presented in two main sections: (i) experiences of microaggressions; (ii) and impact, needs, and coping.

### 3.1. Experiences of Microaggressions

Among the experiences of microaggressions, three themes were identified: (i) types of microaggressions; (ii) perceived motive; and (iii) reaction. An overview of the themes and subthemes identified in this part is shown in Table 1.

#### 3.1.1. Types of Microaggressions

There were eight subthemes identified in this first theme: (i) abusive language; (ii) jokes; (iii) offense; (iv) sexualizing; (v) online; (vi) misgendering and deadnaming; (vii) epistemic exploitation; (viii) denial as a central theme.

##### Abusive Language

Commonly used abusive language included swear words. The most common way to insult participants based on their gender expression was through slurs referring to “being gay”.

Abusive language involved claiming that NBGQ people were attention seekers, that they were not able to choose, or that they had to choose between being a man or a woman. Finley stated that a student in their class explained “non-binary” to other people as “it’s attention seeking people who don’t know what to do with their identity”.

Participants reported aggressors making transphobic comments or trying to make them feel uncomfortable; for example, by asking inappropriate questions. Finley said that their care provider asked them whether they “are a boy or a girl?”. After Finley said that they are non-binary, the care provider responded: “No but do you have a vagina?”. Two participants were told that they had mental problems and should be hospitalized. Ali even stated that people were animalizing them by making barking noises when they passed by.

Participants were blamed for the microaggressions they experienced. Ellis’ mother told them multiple times that “if you get hit because of how you dress, you can’t count on my support”.

Aggressors told participants they did not belong at the current location and that they felt uncomfortable around queer people. This happened especially at gendered toilets but also bars and youth homes.

##### Jokes

Participants frequently experienced jokes about gender diversity or were made fun of. Alex’s identity “was often seen as a joke or something funny”. After Sam shared their gender identity to their teacher, the teacher joked that “then from now on, I am Napoleon”. Finley also mentioned memes on the internet that ridiculed gender diversity in general.

##### Offense

Some participants experienced more offensive forms of microaggressions, such as being threatened, shouted at, catcalled, and harassed. Furthermore, they experienced stalking, disapproving looks, and being avoided because of how they looked. Max shared that “I felt really attacked when I was walking on the street with my partner, someone walked towards us and started shouting things”. Jules stated that, while with their partner, “two men came to us who started pushing our heads together saying kiss for us, kiss for us, but we were sitting next to the water so I was really scared that they would kill us then so that’s the only time I feared for my life”.

##### Sexualizing

Four participants reported being sexualized. Max mentioned being queerbaited, which they described as being flirted with because they looked queer but then being rejected when they got closer. This also included receiving sexualized looks and hand signs because of how they were dressed and aggressors making sex noises. Someone asked Finley, “if I wanted to bang him because I was a guy but not exactly, so he could experiment without really doing it with a real guy”.

##### Online

Alex mentioned that the media can be violent: “Maybe one more thing I consider to be violence is some news articles (…) they always hit me hard, they feel like violence, how they write, and they are using misinformation (…) or they write that these are scientific facts while they are not, and I find it annoying because I know other people will read it and maybe they will not know what’s going on and just believe it”. Participants highlighted the negative consequences of media and the Internet; for example, reading about the bad experiences of other people.

However, the positive aspects of the media and Internet were also mentioned, such as meaningful online friendships that helped them cope. Participants also mentioned a sensation of living in a social media bubble. Charlie mentioned that, “it (what they find on the internet) is targeted, so I will see it (positive queer messages), but people who say oh pronouns, why do we have to put that in our e-mail are not gonna see that because they will block it, so yeah, that’s my own social media bubble”. Alex said: “there was an article, and I thought it was shared to call out a transphobic opinion in the media, so I went to look at the comments to see that other people were also calling it transphobic and then it turned out that I was in the Belgian channel instead of the trans* friendly channel, I just felt safe for a moment and then it turned out I was not in the safe place that I thought I was”.

##### Misgendering and Deadnaming

Deadnaming and misgendering were common experiences. Deadnaming was most commonly experienced with family who did not put in effort or accept the person’s name. Deadnaming went hand in hand with misgendering. Luca said: “my family doesn’t accept it or doesn’t get it and especially the pronouns they and them, they don’t know them, they also don’t wanna try using them, so they don’t call me by my name, and they also don’t use my pronouns”. Misgendering happened frequently and in multiple contexts: from family or friends who were not accepting or misgendering out of habit to fleeting contacts who categorized quickly. Participants had mixed feelings about misgendering. On the one hand, they felt it denied their identity and that it was confirmation they did not pass. Luca said: “it’s like you’re not being seen by your family, who should be close to you”. On the other hand, they often did not react to misgendering, as they understood it was a habit, and it was not the most important part of their identity. Jules shared that, “the whole debate is being reduced to pronouns alone while it is so much more than that, that’s something that I talk about a lot with my parents, that of everything, I think that if they get pronouns wrong, I know I’ve been their child for many years and they have always addressed me in a certain way so pronouns are the last step of my gender identity”.

##### Epistemic Exploitation

Participants mentioned that they were expected to educate others on their identity and issues. To be a “walking search engine”, as Sam phrased it, was experienced as tiring. Sam stated that, “it is really tiring to keep proving, to prove yourself as who you are and to keep explaining the same things. I have even considered creating a PowerPoint or a small leaflet that explains everything and just give it to people”.

##### Denial as a Central Theme

Many microaggressions centered around denial. Participants reported that aggressors denied the existence of NBGQ identities and adhered to binary divisions by stating that only two genders exist or that people who claim to be NBGQ were seeking attention. Finley said that a care provider recommended that they go through a “complete transition” even if they did not want this, which reflects a binary viewpoint.

Six participants said that aggressors had engaged in discussion about their gender identity, denied their gender identity, or denied them as a person. Sam mentioned that, after explaining their identity to their parent for hours, the parent said that, “she will still remain my daughter”. Finley claimed that, because of their gender expression, their parent made comments about their gender identity not being believable. Ellis was even denied as a person: after coming out, their partner told them multiple times that they “had no soul”.

Another form was denial of information about gender diversity; aggressors claimed information to be wrong and inaccurate. Robin mentioned the use of adherence to linguistic correctness to endorse misgendering. In discussions with their friends, “the bottom line usually is that it’s not linguistically correct, ok but what is most important, whether someone feels validated and accepted and also just comfortable, or linguistic correctness”.

Lastly, participants felt denied when aggressors claimed they did not understand how someone could be gender diverse. While talking about mastectomy, Robin’s friend stated that “they couldn’t understand at all why a woman would ever do that”.

#### 3.1.2. Perceived Motive

The second theme consisted of two subthemes; namely, (i) gender expression and (ii) other motives.

##### Gender Expression

All participants referred to gender expression as a perceived motive. Jules experienced more microaggressions after their gender expression changed, and many said they were wearing certain clothing, had just changed their hair, or were looking particularly masculine or feminine when microaggressions occurred. On the one hand, a more queer gender expression or a gender expression typical for the gender opposite to the sex assigned at birth elicited more abusive language, sexualization, and offensiveness.

On the other hand, a gender expression that was in line with expectations for the sex assigned at birth elicited more denial and misgendering. This puts NBGQ youth in a position where, no matter what they do, they may be faced with microaggressions. Alex stated that, “this was one of the first times that I had dressed differently than I normally dress, and this made that, it was clear that my clothes had an impact on how people look at me in a sexual way”, while Robin stated that, “my family and my friends have a certain impression of me and they see a woman and then they stick with that until you actively say well actually not”.

##### Other Motives

Participants referred to intentional motives to do harm. These included the aggressor being transphobic and/or not accepting the participant’s identity. Other motives were that aggressors seemed to get a kick out of it, were inciting each other, or wanted to act tough. According to some, aggressors claimed it was too much effort to not commit microaggressions, such as misgendering and deadnaming. However, only a minority thought that microaggressions were committed on purpose.

More than half of the participants perceived microaggressions as unintentional. They stated that microaggressions, especially misgendering, occurred out of habit, and people did not realize that they were hurting them. Max said: “parents are causing so much violence, but unintended, you know, that, I think often that parents don’t realize what they’re doing and yet they cause so much damage, just by saying something simple or by not thinking about it, they can really hurt me a lot”. Furthermore, incomprehension or having wrong or not enough information were perceived motives. Finley said that microaggressions, such as inappropriate questions, occurred to them because “they thought oh that’s an opportunity to learn”.

#### 3.1.3. Reaction

Three subthemes were identified within the third theme: (i) conversation with aggressor; (ii) no reaction; and (iii) dependent on context.

##### Conversation with Aggressor

Almost all participants engaged in a conversation with the aggressor, which did not always have the desired effect. Ellis said: “I have told them that I am non-binary, and I don’t want to be addressed as she/her but they keep doing it anyway” and Charlie stated that, “when I meet people in real I try to talk about it but it doesn’t always go as well when I, sometimes I try and if I don’t get good reactions I’m discouraged to continue trying”. Robin said that people sometimes reacted well in the conversation about misgendering but continued to commit microaggressions anyway: “I tried to explain it, and she accepts it, that I correct her, but using it herself is just, she accepts when other people do it and that’s fine but using the words herself is still a problem”. Sometimes, aggressors did not react well, as stated by Robin: “that they have a certain opinion about something, and I think that opinion is wrong, or that information is missing, then I find it important I share that information, but then we’re like arguing, or fighting”.

Microaggressions were related to acceptance of gender identity. Max said they “understand that it happens but I just think they sometimes are just not doing their best or something, to accept their child, neighbor or pupil”. Some participants said that they tried to understand why the aggressor had done what they did or put the blame on themselves. Max stated: “I don’t know where people come from or what they have experienced, maybe they have experienced something horrible, it’s possible, but I don’t know that and then I just blame myself, like I shouldn’t have walked there at that hour, I shouldn’t have worn that, I shouldn’t have gone out there wearing this, with those people, and then yeah, I really just blame myself”.

Three participants consoled the aggressor because the aggressor made them feel guilty. Sam said: “she puts herself usually in a position of the victim saying oh but I’m your mother and I’m trying and oh you’re so rude to me but then yeah, I just try to think ok that’s my mom, I’m not gonna be able to change that I think”.

##### No Reaction

Many participants did not react to microaggressions. They alternated between having a conversation with the aggressor and not reacting or shifted from reacting to microaggressions to not reacting. Max said: “usually I reply and then you see in their face that they didn’t realize, and it helps for a while but then, it doesn’t have any long-term effects and then well, I try to still do it but after a while I just stop”. Many reported having no energy or feeling too despondent to react. Max said they, “tried for a while to react, to ask every time oh is it about me and then you notice they stop but after a while it’s just annoying and I had to give up because there is no point and now, I feel that it really asks a lot of energy”. They either blocked off the aggressions, assumed that what happened was normal, looked for distractions, or decided to let people make mistakes. They felt sadness, cried, and hid their pain. Half of the participants said they left the location. When the relationship with the aggressor was already complicated, participants did not react to not complicate things.

Six participants said they did not react because they did not want to explain themselves or educate others. Charlie said: “I have to give an explanation for sure, that’s a requirement, that I can convince them, and I find it so annoying, I just don’t wanna do that”. Max said: “I don’t think it’s my responsibility to educate them because I have the feeling I have done that with so many people already, and they could also just do it themselves” and that “suddenly my uncle shouts at me and says explain the difference between gender and sex, but you know, I know that but why am I being asked to explain that to the whole family and I was just, it was well intended, but it is just incredibly annoying, like I have to teach you but I don’t want to, and if you would just ask if I want to explain I can say no but he just told me to explain, and I did, but it is just so annoying to me”.

Another reason to not react was fearing the aggressor’s reaction or out of safety concerns. They also wanted to avoid a fight or wanted to not appear too extreme. Finley said they, “usually didn’t react because he scared me, he was a really bad person, so I was a little scared of the consequences if I stood up”.

Some participants did not see the severity of the situation at first. Charlie said: “I think I didn’t realize at that moment, maybe even the severity of it, at that moment I just thought, I think I was just laughing with it actually”.

##### Dependent on Context

Whether they reacted depended on the location. Robin said, “I usually don’t react unless I am at an LGBT space”. It was also a result of safety considerations. As Max said, “if it feels bad, I just freeze, then I just, then I just leave, I’m not gonna react to that, I will just take care of my own safety but if I know it’s something small or if it is people that I know I will talk back to them”. Their reaction also depended on whether the aggressor was known or unknown to the person. The effects it had were divided: some reacted more easily when experiencing aggression from a familiar person (such as Max), others when experiencing aggression from a stranger. Sam said: “it hurts me less when the people are unknown to me, I still take it with me but when I don’t know the person, I get angry more easily and I try to explain more, while if I know the person and I’m close to them, well I close myself off, then I don’t know so well what to say”.

### 3.2. Impact, Needs, and Coping

The second part of the analysis centered on the impact and subsequent needs and coping mechanisms resulting from being confronted with microaggressions. Three subthemes were identified: (i) impact; (ii) needs; (iii) and coping. An overview of identified themes and subthemes can be found in Table 2.

#### 3.2.1. Impact

Seven subthemes were identified under the impact theme: (i) gender expression; (ii) fear; (iii) wellbeing; (iv) others; (v) world view; (vi) actions; and (vii) evolution.

##### Gender Expression

Almost all participants felt inhibited in their gender expression because of microaggressions that occurred or the threat of microaggressions (e.g., experiences of others). Luca said: “the first weeks and months I thought ah yeah it’s clearly not ok that I look like this or that I behave like this, so it has, it has inhibited me in that way”.

When asked whether microaggressions had impacted their gender expression, most participants said they had not. However, while talking about gender expression and microaggressions, most participants shared stories of consciously adapting their gender expression.

Most participants adapted their gender expression because of experiences with microaggressions. Some adapted their gender expression to appear more queer to confuse people or to be identified as queer. Sam said they “pay a lot of attention to clothes, what you wear, sometimes I also avoid makeup because for me it isn’t something masculine or feminine but sometimes, I avoid it because I don’t want people to see me as a woman that day”. However, most adapted their gender expression to appear less queer, especially when alone in public. They did this to not have to explain themselves, as this was too exhausting. Jules stated that they, “know it (the gender expression) has an effect and I think that for example I wear more unrevealing clothes to avoid things, or because otherwise it’s too exhausting”. Other reasons to limit expression were to enjoy the protection of passing, because they were discouraged by others, to protect other queer people who were with them, or to avoid suffering. Max said: “I am not 100% happy with it (how they express), of course not, I would like to see it differently, but it’s, I’m not gonna be annoying about it because then I would, I would have to come out again and I just wanna spare myself of that feeling”. Another reason was to appear less queer out of fear of being regarded as inferior, of not being accepted, or of the consequences, such as physical violence or being outed.

Two participants indicated that microaggressions did not change how they expressed their identities, but it made them more aware of their expression. Four participants felt like microaggressions were the price to pay for expressing themselves the way they did. They did not adapt their expression but saw microaggression as something that happened when they appeared as nonconforming. Jules said about gender expression that, “you notice that you have to face the consequences”, and that it is, “a little bit the price you have to pay almost, something you just accept”.

Whether participants chose to adapt their expression depended on the environment (e.g., in a supportive environment or online they were more likely to express themselves), whether they felt comfortable with the people, and whether they were around queer people. Finley explained that, “It makes me not free to present myself how I want in public, I have to be in a very good day to present in an androgynous way if I want to be kept at peace I just go with a sweatshirt and jeans and try to, I can’t really be invisible but try to keep it not really out and proud”.

##### Fear

Participants were afraid to come out or to be open about themselves. Sam experienced, “fear to come out of the closet, actually I still have a lot of fear to be rejected when I say that I am non-binary, I mostly notice with family member I’m very scared to say that and, yeah because even if I do, then there’s all these microaggressions”.

Participants reported fear of being alone in public and of physical violence. Charlie reported, “specific moments that you realize that damn if I would be wearing a skirt here, then it would not have ended well, while I would’ve wanted to wear that”. Finley explained that, “sometimes in places I can get scared because I see a lot of old guys or when I go in a public bathroom I always feel out of place and uncomfortable and there’s still a lot of situations where I don’t really feel safe”.

Sam feared to take medical and legal steps in transition and Ali feared being kicked out of the house.

##### Wellbeing

Many participants reported that microaggression impacted their mental health through frustrations, insecurities, doubts, low self-esteem, exhaustion, anger, sadness, and suicidal thoughts. Jules said: “using swear words or misgendering are so-called microaggressions because ok, it doesn’t threaten you in your physical integrity or something, but I think that when you also look at suicide numbers that in that way it’s more than just a microaggression because it keeps contributing to low self-esteem for example”.

To some, experiencing microaggressions raised awareness that this could actually happen to them. Finley stated that, “it just keeps confusing me that I was in that situation, it feels a little unreal and it’s weird and its, it’s a situation that I heard about online and I heard people talking about it, it’s, it’s weird when I think about it”.

Misgendering and deadnaming, in particular, led to dysphoric feelings, and participants felt their identities were being denied. They also felt inhibited in their identity development. Luca said: “it had the effect on me that I stopped myself to be open about my gender identity to my environment, and to myself” and “I really felt like, like it became clear once again that I shouldn’t be honest about my gender identity, because it wouldn’t be accepted”. Ali mentioned that, “I find it hard to express myself to my parents, let alone in public, and it got so far that I stopped acting manly and that I forced myself to start acting more feminine again”. They also started doubting their gender identity: “first I thought, am I just kidding myself, like am I being serious with myself but yeah, I know that I am still searching so then I sometimes got remarks like oh you’re just confused, you know”. Max even said: “if the right attitudes would’ve been there, it would’ve been just easy to, then I would’ve been able to be non-binary, and that’s not the case now” and “it’s just that, I would just like it if, if it would be like that (that NBGQ identities were accepted by society) but it’s not the case, and I will not mourn it, it’s, it’s ok, well it’s not nice but it’s just ok, I feel like I am being who I am enough to enjoy it but then later I can find more depth (in their identity)”.

Microaggressions elicited feelings of denial. Sam said: “in that moment it was like, well like a piece of my identity was just thrown off the table that was not nice at all because I didn’t feel acknowledged in my identity or in my being, so mostly that, and I remember very well that I, well it was a mix of frustrations and tears and sadness and I thought to myself how could people think like that, and it’s weird, mostly that, and it’s not nice”.

Participants hid their identity more and took decisions between openness and safety. Robin said: “when I notice that someone really has a more aggressive appearance than I will never share my pronouns, then I will just pretend to be a woman completely, I’d rather be misgendered than to get into trouble I think”.

Some stated they were repressing these experiences, such as Max, who said they “have the feeling that I repressed a lot of things that happened because I really, ok it’s not worth remembering, it’s not worth talking about it, I don’t wanna deal with it”.

##### Others

Participants became more distant or less trusting of others. Max stated that, “I feel like, that I still carry these experiences with me, that they are all things that made just a little bit more distant to people, a little bit more careful, and I don’t like that evolution”. Some reported breaking ties with people, relationship problems, or fear of commitment. Max reported that, “I think that my trust has been broken, like the ability to have relationships has suffered”.

Furthermore, they were worried about other queer people’s wellbeing. Max said: “I can handle it pretty well because I’ve been dealing with it (microaggressions) for four years now so it hurts me less, but I would mind a lot if someone else would encounter it that doesn’t have as thick of a skin as I do, that’s the problem”.

The impact of microaggressions depended on the aggressor. Finley said that microaggressions from a parent “really hurt and influenced me for a moment and yeah I took them very personal because he was from my family, and he was supposed to be close to me”.

##### World View

Microaggressions impacted participants’ world views: they lost faith in humanity or did not understand why people would treat them this way. Robin stated that they felt like they were dragged out of their safe bubble: “I have a very good group of friends and I can be myself with my friends because a few of my friends are LGBT themselves and for you it’s the most normal thing in the world, but when you read these comments, you realize again that it’s not the most normal thing in the world for everyone and then you get pulled out of it, you’re hit with reality and that resonates with you, like how is that possible, that it’s still a problem for many”.

##### Actions

Actions that participants undertook after having experienced microaggressions included avoidance and being vigilant in public spaces. Some reported not going to certain places alone or avoiding certain gender expressions. Charlie avoided unknown places and people. A few participants also said they had been warned by their families to be careful.

Participants also compared their experiences to those of others to minimize their experiences. They normalized how microaggressions happened to them and learnt to cope with them. They often blamed themselves. Jules said: “we, as a community, describe many things that are actually more than a microaggression as just a microaggression”. Furthermore, Max stated that, “I encounter many microaggressions but then I hear from friends that they experience more and then I think, is what I encounter this little, it’s like, something weird that, I think it’s something psychological that you, you will minimize your own violence if other people, what they encountered is bigger and I don’t know why that is, but it’s, you have to like prove that it’s real or something”. Sam learnt to handle microaggressions better: “I have learnt, I don’t know, it’s always a reality check when these things happen and then I learn that ok, when it happens again, I can react to it better this way or another way, they’re learning moments in a way but it’s not nice at all”.

##### Evolution

Participants did not initially see the severity of the aggressions or even felt euphoric (e.g., when being misgendered as the gender opposite what was assigned at birth). However, they later realized the severity, which resulted in them being more careful or experiencing feelings of exhaustion. Some participants also indicated that over time they felt less shame and it evolved for the better. Jules said: “the first times that people addressed me as sir or like ah you can’t enter the women’s toilet it was first a feeling of euphoria, like I tricked you, while now I notice that it’s shifting to a form of microaggression for me and how I experience it, initially I liked it because it helped me in my search while now it’s just very tiring”.

#### 3.2.2. Needs

There were two subthemes identified in this theme; namely, (i) validation and (ii) others.

##### Validation

Many participants experienced the need for validation: to be accepted, have their identity confirmed, be seen as an NBGQ person, and feel understood. Jules highlighted the need to accept themselves. Luca said: “I just needed someone in my environment that said that it was fucked up and that I was allowed to look the way I look and that it doesn’t have to mean necessarily that I’m a man or a woman”.

##### Others

Some needed others to seek protection (Charlie: “I thought it was very important, that he, even though I didn’t know the guy, that he could say ah they can’t mess with us, if someone would do something to harm us, yeah I thought that was very important”) or to not feel alone. Max said: “sometimes I need to complain to someone who knows me well, who is queer themselves, that’s nice, that can help and just the feeling that I’m not the only one who experiences this, that’s a nice feeling”.

The most common need was to talk about the experiences to the aggressor or to a therapist or professional. All participants felt the need to talk to friends.

One recurring theme was talking to people (friends or therapists) who were queer or had an affiliation with queerness. They wanted to talk to people who were aware of trans* issues—and to queer friends or therapists specifically—so that they did not have to explain their reality. Jules explained that talking to queer friends helped because, “they are people who experience the same or something, and then you have many ways to talk about it, you can laugh about it, or also share tips, I don’t know, it’s very different than if I would share the story with my parents”. Luca said: “I would like it so much to have a queer therapist, because you, you just have a certain something that you just understand” and “I think what can help is to get in touch with someone that is similar to you, that when I look for help, I can find someone who talks to me who is also queer or also trans, that would be nice”.

#### 3.2.3. Coping

Within this theme, three subthemes were identified: (i) talking; (ii) not coping; and (iii) internal.

##### Talking

The main coping strategy was talking. Two participants needed conversations with their aggressors to cope. Others talked to people who were close to them: parents, partner, and, especially, (online) friends. Ali stated that this helped because, “you get that support, of which you think wow do I even deserve this and then well, you create a nice connection with this person and you know that if something is happening I can share it, and reverse too, and it just feels good, you know that you can be yourself completely with the gender identity you have, you can be yourself completely and can express yourself to this person, and that, it just feels so good”.

Six participants wanted to talk to queer friends as they understood the situation and struggles better. Finley explained that having queer friends was important because of “the shared experience that they have, there are people who have similar experiences to mine and can relate and that makes a lot of things easier of course” and that “it’s really different, we can relate on other levels than I related with other friends, I didn’t lose contact with them but it’s, it’s different in some ways”. Talking to queer friends related to seeking validation. Sam said: “you recognize each other’s stories, so it gives a feeling of connection and it’s very nice to be heard, you get the recognition that you need in that moment, and so it gets easier to let it (microaggressions) go or put them aside”. However, participants also mentioned that they did not always share negative experiences with queer friends because “when I am together with other queer people, I’d rather do fun stuff than not fun stuff” (Max).

Almost all participants talked about microaggressions with their therapists. They did not seek help for these experiences but talked about them with therapists they were already seeing (although Max explained that experiences of microaggression could not be seen separately from the other issues they were facing). Participants indicated a risk of experiencing microaggressions if therapists were not specialized. Jules said: “I could share things about my identity and then they would just say oh that’s interesting, that’s an interesting world you live in, like, my life isn’t a circus you know”. Others said that when therapists were not educated, they were unable to help: “therapists I’ve had are actually examples of people that actually, to whom I had to explain what I experienced instead of that they gave me tools to work with” and “the way they were not prepared to answer my needs was actually really, yeah, really poignant” (Jules). They wanted a therapist who had knowledge about queer issues or was queer themselves because they did not want to explain their identity to the therapist. Max said: “it has been a tough search because I was looking for someone who understands what being queer is because I don’t want to have to explain and I was looking for someone who, well I was really just looking for someone who would understand me, and it didn’t really work out”.

Talking was useful as it helped the participants to find acceptance and to have the feeling that they were not alone. As talking was a prominent need, it seemed that this was often fulfilled.

##### Not Coping

One participant was unable to cope with the experiences as they were in an unsafe environment at home. Various participants repressed these experiences. In particular, it was challenging for them as young people to cope with these experiences, as some lived with their parents or were tied to a certain school environment. Finley said: “it (microaggressions) was coming from pretty much everywhere and I was not leaving it”.

For some, the only way to cope with microaggressions was to leave the unsafe environment. Ali said: “I told the teachers what I thought in a friendly way, saying look It’s a pity that it went this way and that I was not accepted by the students and the teachers, and then I just left” and “it’s a lot easier, now that I’m not at school anymore, it (microaggressions) has diminished a lot, it has gotten a lot better since I left the school”. Finley said: “now it’s better because I just don’t see him (the main aggressor) anymore, I took distance, it’s better”.

##### Internal

Strategies to cope within themselves were attempts to not take it personally and to rationalize and relativize. Some withdrew within themselves to cope. Charlie said: “it (microaggressions) didn’t feel aimed at me, or at least I don’t take it that way, well actually I think I also just really did my best to not take it that way”. Max described that, “I don’t know, then I just take it in or something and then, it’s like something disgusting that you eat and then just digest I think, I don’t bother with it, it’s just swallowing for a moment and then you continue” and stated that they, “go through what happened, if necessary, recording a voice message or something to just make clear what just happened, like that are the worse parts and the less bad parts”. Robin said they, “understand it, that it’s new or unknown to people, it’s starting, now it’s starting to be talked about, it’s starting to appear in the media, which is amazing but a lot of people don’t know it so I understand that for many people it’s not really known and that they are not just gonna start doing it or actively thinking about it themselves if it’s not applicable to them, so in this way I can put it into perspective”.

## 4. Discussion

This research aimed to gain insights into the experiences of NBGQ youth (18–25 years old) with microaggressions and into the subsequent needs, coping mechanisms, and impacts on their lives. Secondly, we wanted to explore how gender expression and nonconformity interact with these experiences.

Participants’ experiences of microaggressions seemed to be centered around feelings of denial and rejection. Common needs were to be validated, accepted, and seen as a NBGQ person and to talk about these experiences with the aggressor or with (queer) friends or therapists. Many engaged in conversations with the aggressors to educate them, although this was a tiring experience and NBGQ often felt too exhausted to do this. Other ways of coping were to try to understand why these experiences had happened or to normalize them. Gender expression interacted with microaggressions, as it was a perceived motive for the microaggressions, and microaggressions also had an impact on gender expression. Often, NBGQ youth adapted their gender expression to either look more or less queer depending on their needs, and if they did not, they thought of microaggressions as the price to pay to express themselves.

Figure 1 shows how the themes brought forward by participants regarding their experiences with microaggressions, their needs and coping mechanisms, and the impacts on their lives and gender expression may relate.

Many experiences were centered around denial: denial of NBGQ identities, of the person’s gender identity, or of them as a person. This was a recurring theme among the different types of microaggressions and closely related to misgendering, which was reported by all participants and can be seen as invalidating and not acknowledging a person’s gender identity. Participants reported feeling denied in their identity and fearing being open about their gender identity and in their gender expression. They felt that many microaggressions denied and rejected their existence and experience and they felt they were not accepted by people. This led to these NBGQ individuals experiencing a fear of rejection and had a direct impact on wellbeing. This was in line with previous research that found that denying trans* people’s identity and experiences, misgendering, and deadnaming are common experiences [28], and that invalidation of identity is common among trans* youth [27]. Although the participants encountered microaggressions that centered around them not being seen or validated as NBGQ people, the findings from the study by Pulice-Farrow et al. [29], which showed that NBGQ people are often doubted with regard to being trans*, were not reflected here. However, this may have been because microaggressions were mostly committed by people who were not queer and may not have had ideas about what being trans* means. In this study, we did not explore if and to what extent participants identified as trans*, so it is possible that not being seen as trans* was not an issue to them as opposed to not being seen as NBGQ.

This study broadens insights regarding the impact of microaggressions. Encountering microaggressions had impacts on the lives of young NBGQ people, who experienced more fear, expressed more vigilance, and employed adaptive behaviors when in public (such as not being alone, adapting expression, avoiding places). This shows that microaggressions are a strong psychosocial stressor, in line with Meyer’s minority stress model [31].

Participants had three main ways of coping. The most important was talking about these experiences, especially with queer (or at least accepting) friends or specialized therapists. Participants indicated that talking to queer people was easier and made them feel understood. Young NBGQ people seemed to be seeking this validation that was denied to them through microaggressions from (queer or accepting) friends who would give them the acceptance they needed. Many stated that they had needs for validation, to be accepted, to have their identity confirmed by others, and to be seen as an NBGQ person. However, there is a risk of encountering microaggressions when talking to an unspecialized therapist.

A second strategy to handle microaggressions was engaging in a conversation with the aggressor. Participants seemed to weigh the importance of explaining how something was hurtful or educating people against the exhaustion of continually explaining their reality. Most participants engaged in conversations with the aggressors but reported that they did not have the desired effect: aggressors did not react well or the effect lasted for a short time. Many participants ended up not reacting because it was too tiring to keep explaining their identity or situation. In turn, the need to explain and to educate others about their identity was seen as a form of violence, and the occurrence of microaggressions and engaging in conversation with the aggressor may be a loop of microaggressions and exhaustion for NBGQ youth. This is in line with epistemic exploitation theory [38]. Participants experienced having to educate others or having to explain themselves as a form of microaggression. In accordance with what Toole [38] suggests, participants explained that educating others took mental energy and emotional labor. These findings are also in line with the study by Fiani and Han [30] that found that non-binary people prefer not to assume the role of educator. It seems that, to avoid this, they turn to queer friends and queer/specialized therapists in order to not have to explain themselves and to get out of this loop by receiving validation. Furthermore, to avoid having to explain or educate, participants often adapted their gender expression.

Another strategy was to rationalize and to try to understand why people commit microaggressions. Participants put the events in perspective and tried to empathize with the aggressor, for whom NBGQ identities were a new topic they had little information about. Some participants even blamed themselves for the microaggressions occurring to them. This may have been especially the case because denial of the identities of young NBGQ people and other microaggressions often occur among people who are close to them, such as family or friends, or people who have a significant impact on their lives, such as teachers or health providers. This reflects the findings in the study by Kiekens et al. [27] that gender minority participants have many experiences of familial microaggressions. In line with attribution theory [42], these coping strategies had an impact on participants’ wellbeing. However, whether young NBGQ people blame themselves for what they do (how they dress, where they walk) or who they are (their identity) should be investigated with more detail in further research, as this influences mental health outcomes [46].

It is striking that many participants perceived microaggressions as unintended, committed because people do not understand NBGQ identities or lack information. It can be questioned whether this is merely a perception or a coping mechanism, such as attempting to not take microaggressions personally. Participants said they took the blame for microaggressions and normalized that they happened. They may not want to see people who are so close or important to them as “bad people”. In this light, it can be discussed whether these “unintentional” microaggressions are as unintended as people claim them to be. All but one participant had conversations with the aggressor, but microaggressions kept occurring to them. This makes the boundary between, for example, misgendering as a habit and not wanting to put effort into stopping it hard to distinguish.

The last strategy adopted was using gender expression. Most participants adapted their gender expression. One option was that they adapted their expression to send the message to people that they are queer. This way, they could try to nonverbally explain themselves as a NBGQ person to possible aggressors. This was in line with the study by Fiani and Han [30], who found that passing is challenging. The other option was to adapt their expression to be less visibly queer to avoid microaggressions. This was partly due to the weighing of safety concerns, as also found in the study by Fiani and Han [30]. However, an additional consideration of expression versus exhaustion (because of having to explain and encountering microaggressions) was found. Participants weighed whether they were too exhausted to deal with microaggressions or not.

If participants did not adapt their gender expression, they experienced microaggressions as the price they had to pay for expressing themselves, which in turn can contribute to normalizing microaggressions. It indeed seems that gender nonconformity is an incentive and that visible forms of “being different” lead to more stressful situations, as research [21,22,23,24,25,26,27,28,29,30,31,32] has suggested. Gender nonconformity was perceived as an incentive for microaggressions, which is in line with findings that gender nonconformity is an incentive for general violence [34]. While the findings of the FRA [13] show that, for trans* people, openness about gender identity is related to the amount of violence they experience, this study found that microaggressions people experience also influence their openness about their gender identity.

## 5. Conclusions

This study broadens research on microaggressions with a trans* and NBGQ lens. Specifically, it provides an in-depth look at the experiences of microaggressions of a more specific subgroup rather than looking at microaggressions and the LGBTQ or trans* community as a whole [24,26]. This research combined different aspects of research on microaggressions, gender expression, and epistemic exploitation and tried to find how they are interconnected.

The study found that experiences of microaggressions among NBGQ youth were centered around denial of their identity or NBGQ identities in general. They felt rejected in their existence and experience and felt unaccepted by people. To cope, they talked to queer or accepting friends or therapists to seek validation. They also tried engaging in conversations with the aggressor, although this did not have the desired result. The young NBGQ people weighed the need to educate and explain to the aggressor against educating being experienced as tiring and exhausting. Another strategy was to rationalize and try to understand the aggressor. This led to the young NBGQ people perceiving microaggressions as unintentional and engaging in self-blame or normalization. Lastly, microaggressions and gender expression interact. On the one hand, the young NBGQ people adapted their gender expression to appear more queer to “explain” their identity to people; on the other hand, they adapted their gender expression to appear less queer to avoid rejection. The importance of expressing themselves was weighed against the exhaustion of dealing with microaggressions. When NBGQ youth did not adapt their gender expression, they experienced microaggressions as “the price to pay” for expressing themselves.

One limitation of this study was that it had a relatively small sample size, and it is not possible to generalize these results. However, it gives indications of which aspects play a role and what is important to NBGQ youth when it comes to microaggressions. These different aspects and how they influence each other can be further researched. As this research was explorative, further studies can research this more specifically and in detail. The topics found in this explorative study can serve as a guide for questionnaires or interview guides in further studies.

While focusing on a small age category may give more reliable results about this specific group, this study cannot speak for the experiences of all NBGQ people, and research will need to map out experiences of older NBGQ people.

Another limitation is that a certain profile of NBGQ youth was overrepresented. All participants were white and able-bodied, and most reported they had studied or were studying. All of them were open about their identity to at least one person. Almost all mentioned having a network of queer friends, online or in real life, with whom they could be themselves. As this is a useful coping mechanism, it would be valuable to explore how NBGQ people who are not out or do not have a network of queer friends cope and how they can be assisted.

As this was an initial exploration of young NBGQ people’s experiences with microaggressions, further research is needed to confirm every aspect of how these experiences relate in detail. For example, a future study could investigate how the relation with the aggressor impacts the experience of microaggressions, and if this is related to NBGQ people experiencing microaggressions as unintentional or empathizing with the aggressor. Furthermore, the impact of microaggressions through the media and the Internet could be studied, as this aspect could only be briefly touched upon here. Though it is more challenging, it may be interesting to not only look at perceived motives but to study the attitudes and experiences of the friends and family of NBGQ people with whom they have these conversations.

It’s important to note that participants categorized microaggressions mostly as unintentional. As Sue posits that unintentional forms of microaggressions can be dangerous as they go unseen [25] (participants often did not realize the severity at first), and they are more pervasive, future research should focus on this aspect in more depth.

Lastly, as one prominent need of NBGQ people is to talk to a therapist who is queer or specialized in queer issues, one recommendation is that efforts must be made to ensure the availability and accessibility of specialized therapists.

## Figures and Tables

**Figure 1 healthcare-11-00742-f001:**
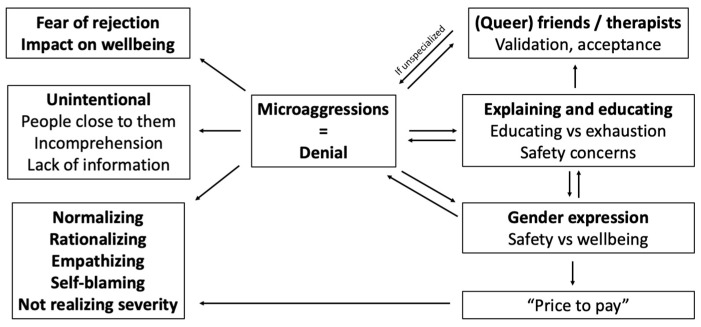
Experiences of microaggressions among NGBQ youth. This figure summarizes the main experiences of NBGQ youth with microaggressions and how different elements of these experiences relate.

**Table 1 healthcare-11-00742-t001:** Experiences of microaggressions: themes and subthemes.

Themes	Subthemes
Types of microaggressions	Abusive language
	Jokes
	Offense Sexualizing Online Misgendering and deadnaming Epistemic exploitation Denial as a central theme
Perceived motive	Gender expression
	Other motives
Reaction	Conversation with aggressor
	No reaction
	Dependent on context

**Table 2 healthcare-11-00742-t002:** Impact, needs, and coping: themes and subthemes.

Themes	Subthemes
Impact	Gender expression
	Fear
	Wellbeing Others World view Actions Evolution
Needs	Validation
	Others
Coping	Talking
	Not coping
	Internal

## Data Availability

The data presented in this study are available on request from the corresponding author. The data are not publicly available to ensure anonymity for the participants.

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
