# Peer review of "The Price to Pay for Being Yourself: Experiences of Microaggressions among Non-Binary and Genderqueer (NBGQ) Youth"

_healthcare, 2023, doi:10.3390/healthcare11050742_

Round 1

Reviewer 1 Report

The essay is very well researched, organized and presented. I particularly value the author's self-awareness about the possible shortcomings of their investigation. I encourage them to amplify the study by including a wider variety of people in their subsequent research, but the article as it is is focussed and neat.

I have found a few minor spelling mistakes:

-Closing parenthesis missing on line 109.

-"To" missing on line 138-

-Number 12 is not bracketted or is misplaced on line 401.

-Space missing on line 451, "expectingfor"

-On line 563, an "s" missing: "identified a queer"

-Space missing on line 717, "othersto."

Otherwise, I find the work most relevant and interesting to read.

Author Response

Please see attachment for response to the review report.

Reviewer 2 Report

The article is very interesting as it deals with a very timely topic -microagressions for a specific LGBT group- non-binary gender groups- that is  defined in respect to other LGBT people, opening a new field of a nlysis. This is especially original as non-binary identification is a quite new phenomenon. However, the overlapping of subjective and objective dimensions needs further elaboration to clarify the effective contribution of the collected data (a small sample) to further research on the topic.  Some concepts (including the same non-binary) need to be more specified and developed. A doubt about the definition of adolescents (18-25) while 18-25 includes young men -the article  defines adolescents between 18-25 years old- . This age group18-25 does include young adults, but the article doesn't make any distinction between adolescents and young adults. The definition of microagressions needs as well to be specified: references are good, but the approach shifts from subjective perceptions to objective facts, symbolic violence and physical violence, etc...The Belgian context is not clearly described in respect to the issues raised in the article and more in general.  The article is explicit on the fact that data cannot be generalised because of a small sample, but a more in depth analysis and a more precise description of the sample might have partly contributed to offer more general conclusions.

Author Response

(The authors gave the same response as above.)

Reviewer 3 Report

The article addresses a relevant topic. It integrates a qualitative study where the discourses of 10 participants are explored with the aims of “gain insights into experiences of NBGQ youth (18-25 years old) of microaggressions, and into subsequent needs, coping mechanisms and impact on their lives. Secondly, it wanted to explore how gender expression and nonconformity interact with these experiences.” The discourses of these NBGQ youth were explored through individual semi-structured interviews, an appropriate technique for this purpose, and were analysed using the NVivo and thematic analysis methodology. I therefore consider that this article should be accepted for publication. However, in order to do so, major changes will have to be made, especially in the Materials and Methods, Results and Discussion.

I really appreciated the theoretical framework that started from the general/international to the particular/country/Belgium and ended very well with the research objectives.

But I didn't feel the same about the Materials and Methods section. To make this section clearer, I think it should be divided in the most “traditional” form, with the 4 usual sections: Participants, Procedure, Materials/Instruments and Data analysis, or Analytical strategy. This way, it would also be easier to replicate the study.

Participants - In this section (and not at the beginning of the Results section as is done), participants must be characterized.

Procedure - In this section, you must ensure that all information related to the procedure is mentioned, that is, whether an opinion was sought from the Ethics Committee of one of the authors' institutions; how you reached the participants; whether, prior to the interviews, the 10 participants read an informed consent; if they were informed about the objectives of the study; where the interview was carried out; how long they took (it should be indicated how long the shortest and longest interviews took; or the average of all interviews); whether the interviews was recorded with their consent; whether their anonymity was assured; whether the interviews were transcribed in full, and so on.

Material/Instrument - This section should contain information about all the material or instrument used in the study. For example: what is the origin of the interview guide, or how was it constructed (for example, from the theoretical framework?).

Data Analysis - Here, it is not enough to say that thematic analysis was used. It is important to add at least what this method consists of; what exactly it allows to do or identify; how the analysis was done; who did the data analysis, was it done by one author or by all authors? For example, is it Braun and Clarke's thematic analysis? If yes, it is necessary to mention it, say what this methodology consists of and highlight the 6 phases that it encompasses.

As you may know, in qualitative research articles or reports, it is always good to have a look at the COREQ (Consolidated criteria for Reporting Qualitative research) Checklist, a checklist of items that should be included in reports of qualitative research.

As the results are presented in the manuscript, they are very difficult for the reader to follow. At the beginning of the presentation of the results, it would be better and clearer if you started by presenting the themes by also inserting a table with the identified themes and sub-themes. See, for example, how did the authors of the following article publish in Social Sciences:

Ferreira, R. A., & Santos M. H. (2022). Gender and ethnicity: The role of successful women in promoting equality and social change. Social Sciences, 11(7), 299. https://doi.org/10.3390/socsci11070299

Note that the authors begin by presenting their analysis on the themes and sub-themes identified and only then presented the excerpts from the interviews to illustrate them. This makes the presentation of the results much better and clearer. I suggest that you do the same. Also, instead of presenting the discourses of each of the participants, you should first present each of the sub-themes and then present the interview excerpts that best illustrate them.

Regarding the Discussion section, much of the discussion is not really a discussion of results in light of theory, only the last 4 paragraphs do that.

To finish, I have a question. If the article was written by several authors, why is it sometimes written in the first person?

Author Response

(The authors gave the same response as above.)

Round 2

Reviewer 2 Report

I thank the authors for the important work of revision they made and I consider that the article can be published in this new version.